# The role of EMODnet Chemistry in the European challenge for Good Environmental Status

Matteo Vinci[1], Alessandra Giorgetti[1], Marina Lipizer [1]

[1] OGS (Istituto Nazionale di Oceanografia e di Geofisica Sperimentale) Borgo Grotta Gigante 42/C - 34010 - Sgonico (TS) - Italy

*Correspondence to: Matteo Vinci mvinci@inogs.it, Alessandra Giorgetti agiorgetti@inogs.it, Marina Lipizer mlipizer@inogs.it*

**Abstract**: the European Union set the ambitious objective to reach within 2020 the goal of Good Environmental Status. The Marine Strategy Framework Directive (2008) represents the legislative framework that drives Member States efforts to reach it. The Integrated Maritime Policy supported the need to provide a European knowledge base able to drive sustainable development by launching in 2009 a new European Marine Observation and Data Network (EMODnet). Through a stepwise approach, EMODnet Chemistry aims to provide high quality marine environmental data and related products at the scale of regions and sub-regions defined by the Marine Strategy Framework Directive. The Chemistry Lot takes advantage and further develops the SeaDataNet pan-European infrastructure and the distributed approach, linking together a network of more than 100 National Oceanographic Data Centres providing data from more than 500 data originators. The close interaction with EEA, RSCs, ICES and EMODnet-MSFD coordination group allows to assess the most proper set of information necessary for the MSFD process. EMODnet Chemistry provides aggregated and validated regional data collections for nutrients, dissolved gasses, chlorophyll and contaminants, properly visualised with OGC WMS and WPS viewing services. Concentration maps with 10-year moving window from 1960 to 2014, by season and for selected vertical layers are computed and made available.

**Keywords**. Marine chemistry; Europe; Marine Strategy Framework Directive; Good Environmental Status; Eutrophication; Contaminants;

## 1. Introduction

The European Union has set the ambitious objective to reach within 2020 the goal of Good Environmental Status (GES) for our oceans and seas. The challenge consists in facing the environmental degradation caused by years of unsustainable and inefficient growth model. The Marine Strategy Framework Directive (MSFD, European Commission 2008) adopted in 2008, with its eleven descriptors and related indicators, represents the legislative framework and the backbone of this work. MSFD defines the GES in Article 3 as: “The environmental status of marine waters where these provide ecologically diverse and dynamic oceans and seas which are clean, healthy and productive”. GES means that the different uses made of the marine resources are conducted at a sustainable level, ensuring their continuity for future generations. Going more in detail for the restoration and the safeguard of this status, the ecosystems (including their hydro-morphological, physical and chemical conditions) should be fully functioning and resilient to human-induced environmental change, the decline of biodiversity caused by human activities should be prevented and biodiversity is protected. The human activities (introducing substances and energy) shouldn’t cause pollution effects and noise from human activities should be compatible with the marine environment and its ecosystems.

This process is at the moment in the half way between the adoption (2008) and its deadline (2020). The main features of this strategy are: the ecosystem approach (to provide an integrated evaluation of the activities affecting our seas) and the common efforts required by the Member States for the cooperation between neighbouring countries.

Some efforts have already been undertaken by European Member States which provided in 2012 the initial assessment on the state of the environment of the national marine waters. This assessment reported on environmental status determined in a holistic way, according the 11 descriptors, and on the objectives and targets to reach GES following the articles 8, 9 and 10 of the MSFD.

The results of the first phase allowed to recognize gaps and needs in data availability, large heterogeneity of methodological approaches to report information and spatial inconsistency within Member States regarding coastal – offshore data. These outcomes clearly indicated that more efforts are urgently needed if the EU wants to reach its goal. More has to be done on the cooperation side and especially on the integration between Member States and Regional Sea Conventions (RSC). The report from the Commission on the first phase of implementation of MSFD indicates a high level of heterogeneity among Member States reports and in several cases poor data availability and accessibility (Dupont et al., 2014; Palialexis et al., 2014)

As a consequence, evaluation at higher level (Regional and EU) is difficult to perform. This first phase of MSFD implementation has somehow brought Europe one step closer to the ecosystem approach. However, the recognized gaps in data and information, the high heterogeneity in assessment approaches should guide the stakeholders involved in MSFD implementation to develop a more homogeneous approach. In view of the revision of the assessment in 2018, several efforts are required to overcome the shortcomings identified in the first reporting phase. Going more in detail, the actions should be focused on different aspects like: revised criteria for GES, methodological standards and standardised methods for monitoring, assessment and data availability, implementation of integrated information systems at regional and EU level.

In the field of marine research, during the last decades several oceanographic data management initiatives faced the challenge of data availability, interoperability and resilience at Pan-European level (EU MAST MTP II MATER 1996-1999, EU MAST-INCO MEDAR 1999-2001, FP6 SeaDataNet 2006-2011, FP7 SeaDataNet2 2011-2015).

Interoperability is defined as “the ability of a system to work with or use the parts of another system”, while resilience is defined as "the ability of a system to cope with change". The translation of these principles in the oceanographic data management consists in the development of a long life system able to easily interact with other systems. As example the adoption of common formats for data and metadata and a system of common vocabularies ensure that the network of involved persons is working in a homogeneous environment from the syntactic and semantic point of view (speaking a common language). The resilience is safeguarded by metadata and quality flags that provide clear knowledge of which kind of information the users are handling even long time after the data measurement (e.g. use of historical data for time series studies).

Since 2007, the Directive establishing an Infrastructure for Spatial Information in the European Community (INSPIRE, 2007/2/EC) has been the driving principle to ensure that the European spatial data infrastructures are compatible and usable in a transnational context. The Directive requires that common implementing rules are adopted for the organisation, accessibility and sharing of spatial information with a focus to the implementation of interoperability of spatial data sets and services. Marine data management communities, developed in the framework of European initiatives such as the above mentioned MATER (1996-1999) and MEDAR (1999-2001) that converged later in the SeaDataNet (2006-2015) experience, faced the challenge to provide access to the huge amount of already existing but

fragmented and inaccessible data collected by EU oceanographic institutes. This was done developing a system able to collect, standardise, quality control and share the information, taking into proper account the data policies.

The simple but efficient idea was the active collection of the EU oceanographic data at national level carried out by a network of National Oceanographic Data Centres (NODCs). The collection of those data was done in direct communication with the data originators to ensure the best set of measured data and related metadata. Metadata, that are all the information needed to describe exhaustively the data, reply to a set of basic but fundamental questions like: who, where, when, what and how about the collected information. For this reason they are key elements to enable efficient browsing and discovering.

Between the data collection and sharing, the crucial steps to ensure interoperability and reliability consist in standardization and quality control.

The standardization is done at two main levels by following the interoperability principles provided by INSPIRE: syntactic and semantic. The first is done providing common formats for the files providing metadata and data (XML ISO, ascii). The second is done by means of a set of common vocabularies that let to "use the same language" to describe data and metadata over time, different projects and nationality.

The quality control procedures provide the necessary labelling to complete the harvested information with the evaluation of their reliability.

Finally the registered users can access the needed information according to data access and usage policies defined in agreement with the originators.

In order to extend this approach to different disciplines of the marine environment, at EU level, the Directorate-General for Maritime Affairs and Fisheries (DG-MARE) launched since 2009 a set of thematic contracts to establish a European Marine Observation and Data Network (EMODnet).The aim of the initiative was to improve the availability of high quality marine environmental data at the scale of regions and sub-regions of the Marine Strategy Framework Directive, to build a knowledge base that can assist in the implementation of marine policies and drive sustainable development. The EMODnet Lots with their infrastructure could play a central role specifically for countries where the Regional Sea Conventions are less mature to support the need of qualified and standard information at national, regional and bigger scale.

## 2. Background

A pilot project was launched by DG-MARE in 2009 to create the components of the European Marine Observation and Data Network (so called ur-EMODnet), as proposed in the EU Green Paper on Future Maritime Policy (European Commission 2006), consisting in six thematic data portals managing data on bathymetry, marine geology, chemistry, biology, seabed habitats, and physical oceanography. Based on the successful experience of the SeaDataNet (SDN) project (7th Framework Program), EMODnet Chemistry adopted its approach (Vinci et al., 2013). The principle was to take advantage of its efficient and distributed infrastructure for the management of data deriving from in situ and remote observation of seas and oceans. This infrastructure can be considered a European de-facto standard, as it already involves around 100 institutes (nodes) from 35 countries and is adopted and continuously adapted according to specific requirements for chemical data management.

SeaDataNet is actively involved in the development of standards that follow the INSPIRE implementing rules to ensure interoperability such as:

- Common metadata standards based on the Extensible Mark-up Language (XML), based on ISO 19115/19139 schema;
- Standard data transport formats Ocean Data View (ODV) ASCII, MEDATLAS and NetCDF (CF);
- Common quality control methods and quality flag scale;
- Common Vocabulary Web services, used to mark-up metadata and data, covering a broad spectrum of disciplines and governed by an international board (SeaVox);
- SOAP Web services for various communication tasks;
- Open Geospatial Consortium (OGC) compliant services (Web Map Service, Web Feature Service, Web Processing Services) for viewing services of data products.

The partnership involved a subgroup of the SeaDataNet network of National Oceanographic Data Centres (NODCs) with specific experience in data collection, in data analyses, validation, and creation of products and in the technical partners who further developed SDN infrastructure. The Chemistry Pilot project was focused on the collection and management of data on some chemical parameters relevant for the MSDF (contaminants and fertilisers), in three matrices (sediment, seawater and biota) and in three areas of interest: the North Sea, the Black Sea and some spots in the Mediterranean Sea.

The comparison of the harvested data between sea basins highlighted a highly heterogeneous situation according to the different parameters. Data distribution consisted, on one hand, in coastal time series stations monitored at regular temporal scale, on the other, in data homogenously distributed at basins level, but discontinuously in time. Furthermore, high heterogeneity in data managed resulted in the different sampling and analytical protocols adopted, as well as in the different target species. As a last step of the pilot project data visualizations were provided as interpolated maps when data were homogeneously distributed in time and space and as time series plots to allow visualization of data with fragmented spatial coverage. The viewing products were made available on the dedicated web portal in OGC compliant format (WMS layers).

## 3. EMODnet

The positive outcomes from the pilot project confirmed the interest in the further development of a marine observation infrastructure able to provide data and knowledge required to support the development of marine economy whilst supporting environmental protection needs, as underlined in the Green Paper Marine Knowledge 2020 (European Commission 2012). The new phase includes data collection for all European sea-basins: the Baltic Sea, the North-East Atlantic Ocean, the Mediterranean Sea and the Black Sea and involves 46 partners (Fig.1), both from research institutes and national monitoring agencies.

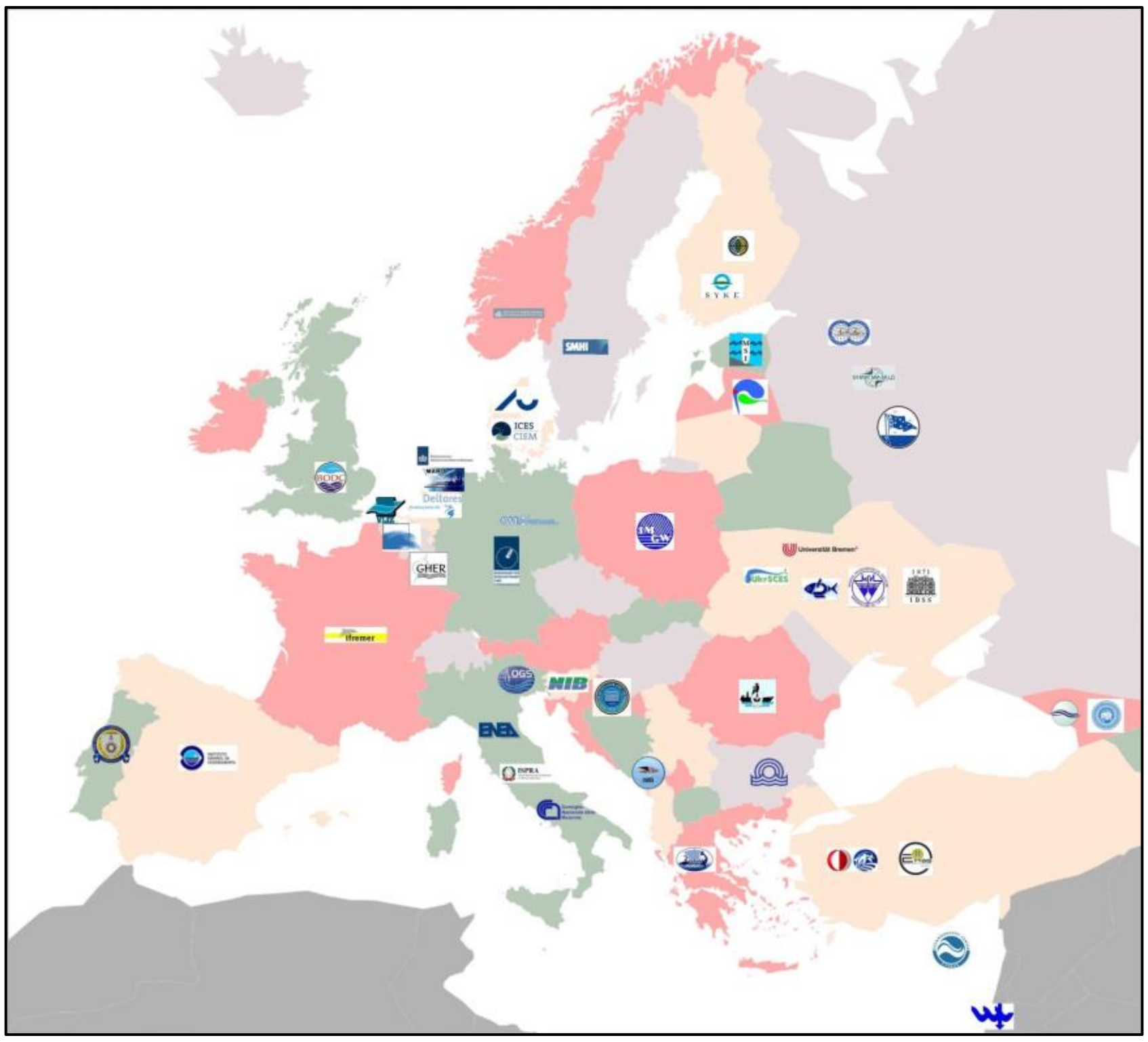

**Fig. 1: Geographic coverage of EMODnet Chemistry partnership. Logos indicate the nationality of the partner institutes.**

Data managed by EMODnet now include also silicates, chlorophyll, partial pressures of dissolved gases (oxygen and carbon dioxide), plastics (polyethylene, polypropylene) and acidity (pH, pCO2, Total Inorganic Carbon, alkalinity).
Data collection and product generation for all European basins is carried out by 5 Regional leaders, responsible for the North Sea, the Baltic Sea, the Atlantic, the Mediterranean, and the Black Sea.
In order to better tune EMODnet efforts for the requirements of the MSFD, several initiatives have been carried out to strengthen the dialogue with the Regional Sea Conventions and the Marine Observation and Data Expert Group (MODEG) and a MSFD – EMODnet coordination group involving Regional Sea conventions, Member States and relevant stakeholders has been established jointly by DG Mare and DG Environment. Besides, regular meetings with INSPIRE implementing groups are organized to discuss on the most feasible and useful products and services to provide.

## 4. Data collection and Access

Data harvesting is a fundamental activity of EMODnet and it is carried out by the network of NODCs that supervise the national availability of research and environmental monitoring data, provided respectively by research institutes and environmental agencies (Fig.2). NODCs maintain regular contact with data originators collecting and enriching data with the best set of relevant metadata to ensure the reliability of the information. NODCs are also responsible for the first quality control of data, flagged with quality information.

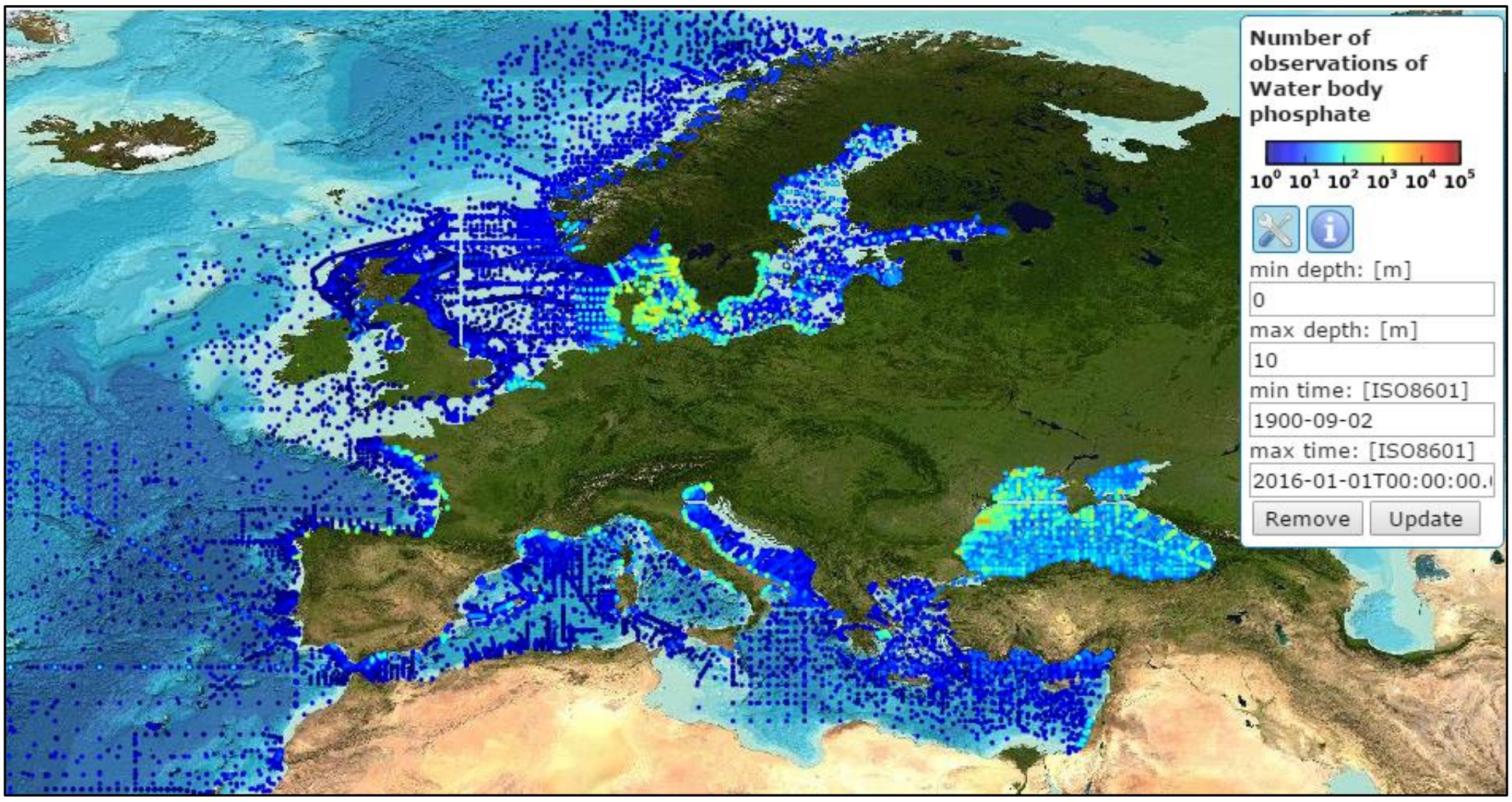

**Fig. 2: Density and distribution of water body phosphate harvested stations**

Data access is regulated by a data policy (defined in agreement with data originators) which aims to establish a balance between the right of the originator to get proper acknowledgment for data acquisition, and the need for open access through free and unrestricted exchange of data, meta-data and data products. The analysis of data policies for EMODnet Chemistry data shows differences between data access restrictions for nutrients and contaminants (Fig. 3).

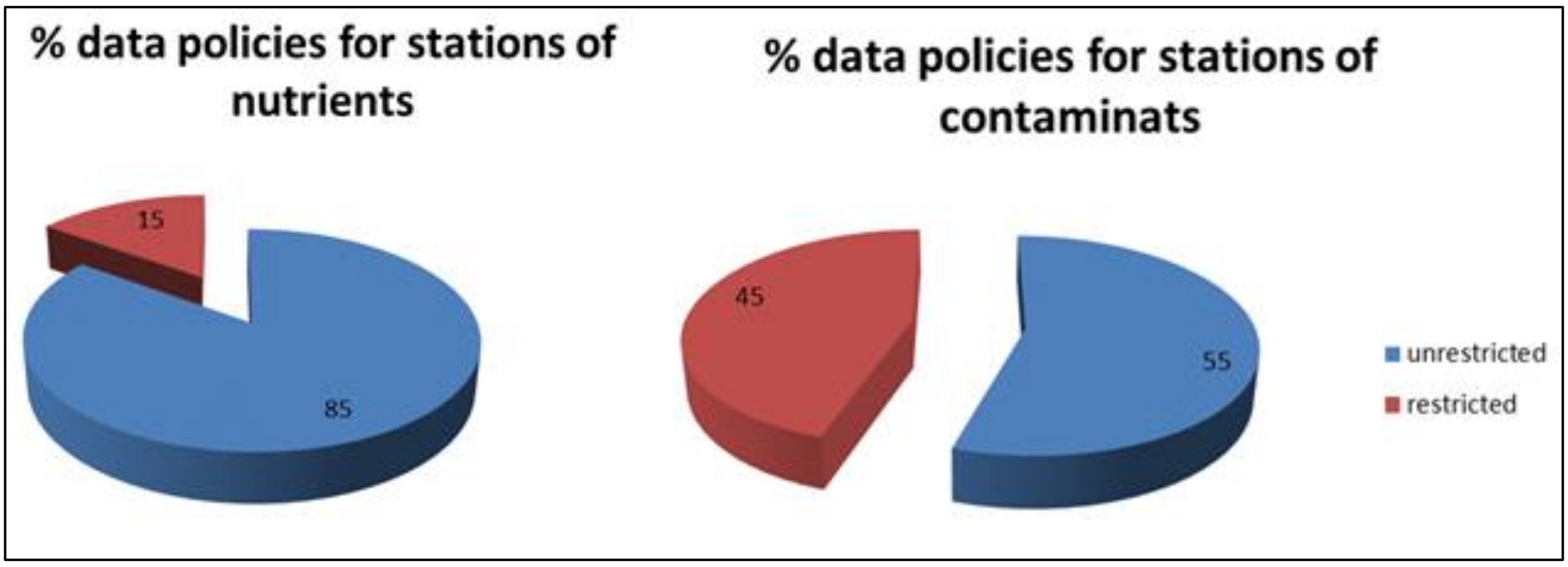

**Fig. 3: Data policy for nutrients and contaminant data**

Data requests from registered users are handled by NODCs through a data policy management system. Unrestricted data are freely available while restricted ones need negotiation with data originators. This kind of filter on data access is an effective way to establish contacts and trust between data originators and data management centres, ensuring correct acknowledgement, which ultimately encourages data sharing.

By maximizing the availability of data to a larger community, SeaDataNet promotes the use of these data, thereby ensuring that their maximum value can be realized and thus contribute to increase knowledge of the marine

environment. Fig. 4 shows temporal distribution of nutrient data, spanning from 1900 to 2016; table 1 shows the number of stations for parameters with more data available.

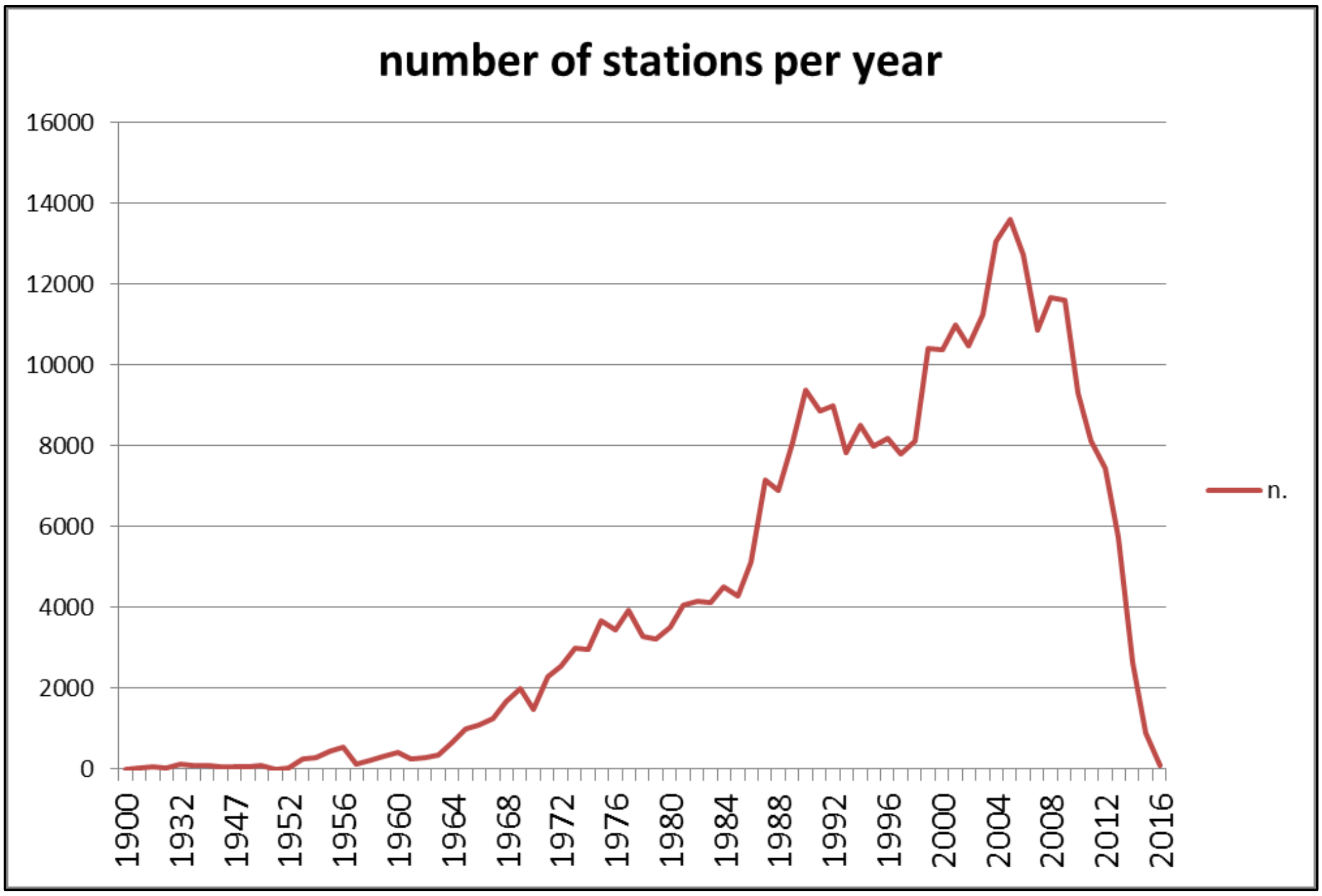

**Fig. 4: Temporal distribution of nutrient data, spanning from 1900 to 2016 (counting 90 profiles in the current year; updates May 2016).**

| Parameter | n. |
|---|---|
| Phosphate concentration parameters in the water column | 305896 |
| Nitrate concentration parameters in the water column | 262378 |
| Silicate concentration parameters in the water column | 245755 |
| Dissolved oxygen parameters in the water column | 198357 |
| Ammonium and ammonia concentration parameters in water bodies | 188666 |
| Nitrite concentration parameters in the water column | 181642 |
| Salinity of the water column | 151969 |
| Chlorophyll pigment concentrations in water bodies | 145374 |

**Table 1 number of stations for the parameters with more data available**

## 5. Data Quality

The quality of the data is a key issue when merging heterogeneous data coming from different sources, periods and geographic areas. Within EMODnet chemistry community, commonly agreed and standardized data quality control (QC) protocols have been defined (Holdsworth, 2010) to guarantee consistency among comprehensive databases which

include data from different and/or unknown origin and covering long time periods. As a first step, the data are checked and completed by collators with a standard set of metadata that provide the basic information necessary for their long term use. Afterwards, data undergo a validation loop which consists in several validation steps. The first is done by data collators, prior to the inclusion in the decentralized infrastructure and the second step, which consists in regional quality control, is performed at regional scale on aggregated datasets. The first quality controls (QC) ensure that position and time of data are realistic and compare measurements with broad ranges and specific regional ranges. Whenever available, data are also compared with climatology. As a result of the first QC step, all data are archived with a quality flag value that provides information about their reliability.

At this point, data aggregation and regional quality control are performed at regional scale, following a common protocol. Data aggregation is done with the objective to unify the various analytic terms into a unique aggregated term with conversion to a unique measurement unit. The ODV software has a built-in aggregation procedure applying a number of business rules like possible units conversions. (Lowry R. et al., 2013)

The main goal of this activity is to obtain a harmonized dataset (e.g. a unique dataset of phosphate concentration in the water column starting from different datasets of phosphate concentration expressed with different units) that could be used to generate homogeneous data products. The results of the regional quality control are sent to the data collators (NODCs) to correct errors or anomalies in the original copy of the data available in the EMODnet infrastructure. This feedback loop guarantees data quality upgrade (Fig.5).

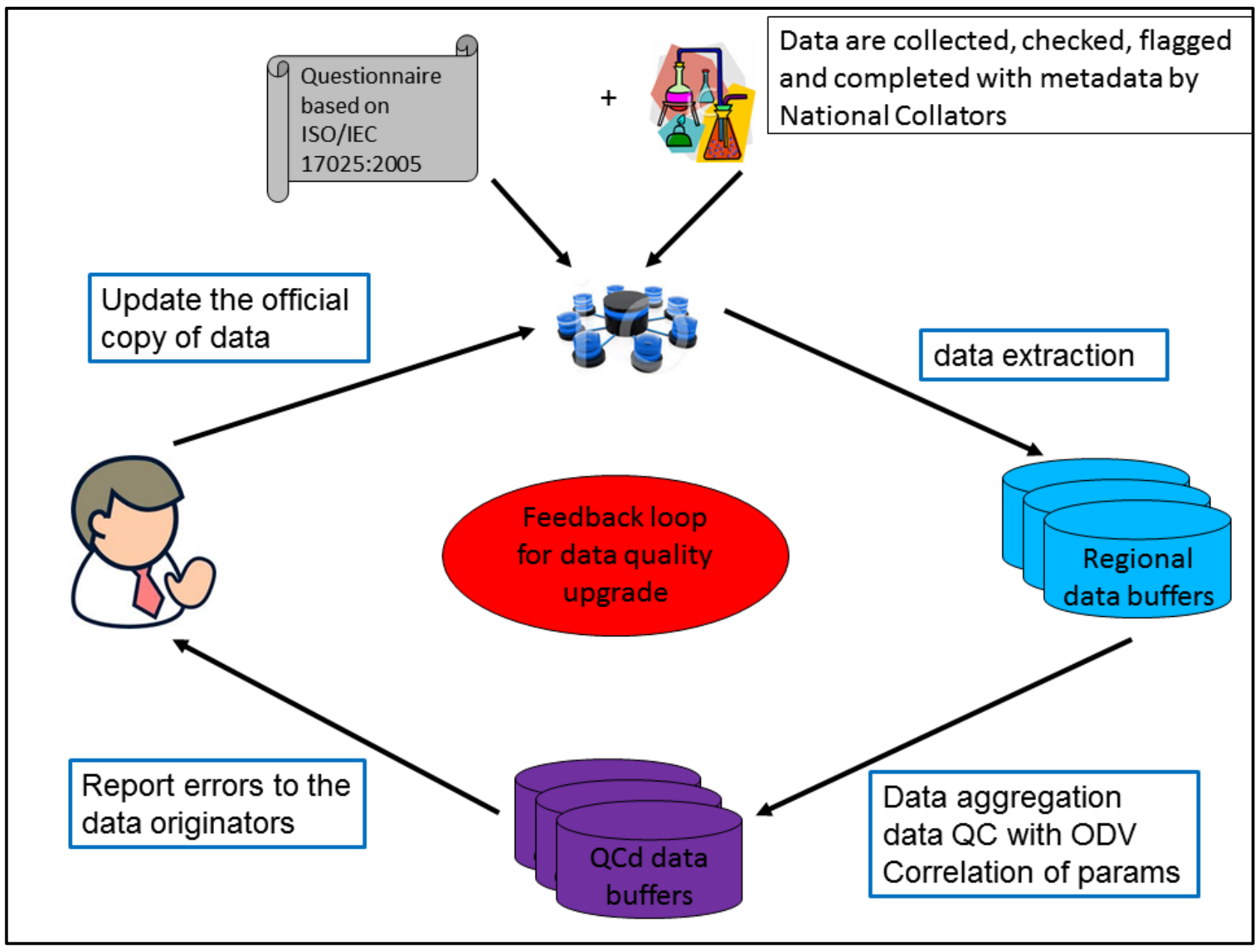

**Fig. 5: Data validation loop**

To improve and homogenize the quality control procedures and standards adopted (at least at regional level), a quality control survey has been carried out within EMODnet Chemistry community, in order to collect the best practices in data validation and highlight gaps of the different institutes involved (Vinci et al., 2015).

## 6. Data products

In order to accomplish the Marine Strategy Framework Directive requirements, EMODnet Chemistry developed products suitable to visualise the time evolution of a selected group of measurements and to calculate spatially

distributed data products specifically relevant for MSFD descriptor 5 (eutrophication), 8 (chemical pollution), and 9 (contaminants in seafood) as typically done with satellite data (Colella et al., 2016; Gohin et al., 2008)
The interpolated maps have been produced with the Variational Inverse Method (VIM; Brasseur et al., 1996), using the software DIVA (Data-Interpolating Variational Analysis; Troupin et al., 2010). DIVA is an appropriate numerical implementation of VIM suitable for oceanographic data spatial analysis as it is designed to obtain a gridded field from the availability of non-uniformly distributed observations (Barth et al., 2010; Troupin et al., 2012).
Interpolated maps are now generated, mainly for nutrients, with 10 years moving window in order to find a balance between the duration of the environmental evaluation cycle for Member States (to provide maps with a time frame near to the 6 years process of the Member States evaluation) and the number of years that guarantee a sufficient data coverage.
An example of a visualization useful for the assessment of eutrophication and in particular of nutrient concentration in the water column is presented in Fig. 6, which displays surface distribution of phosphate concentration in spring for the decade 2003-2013 (centered in 2008).

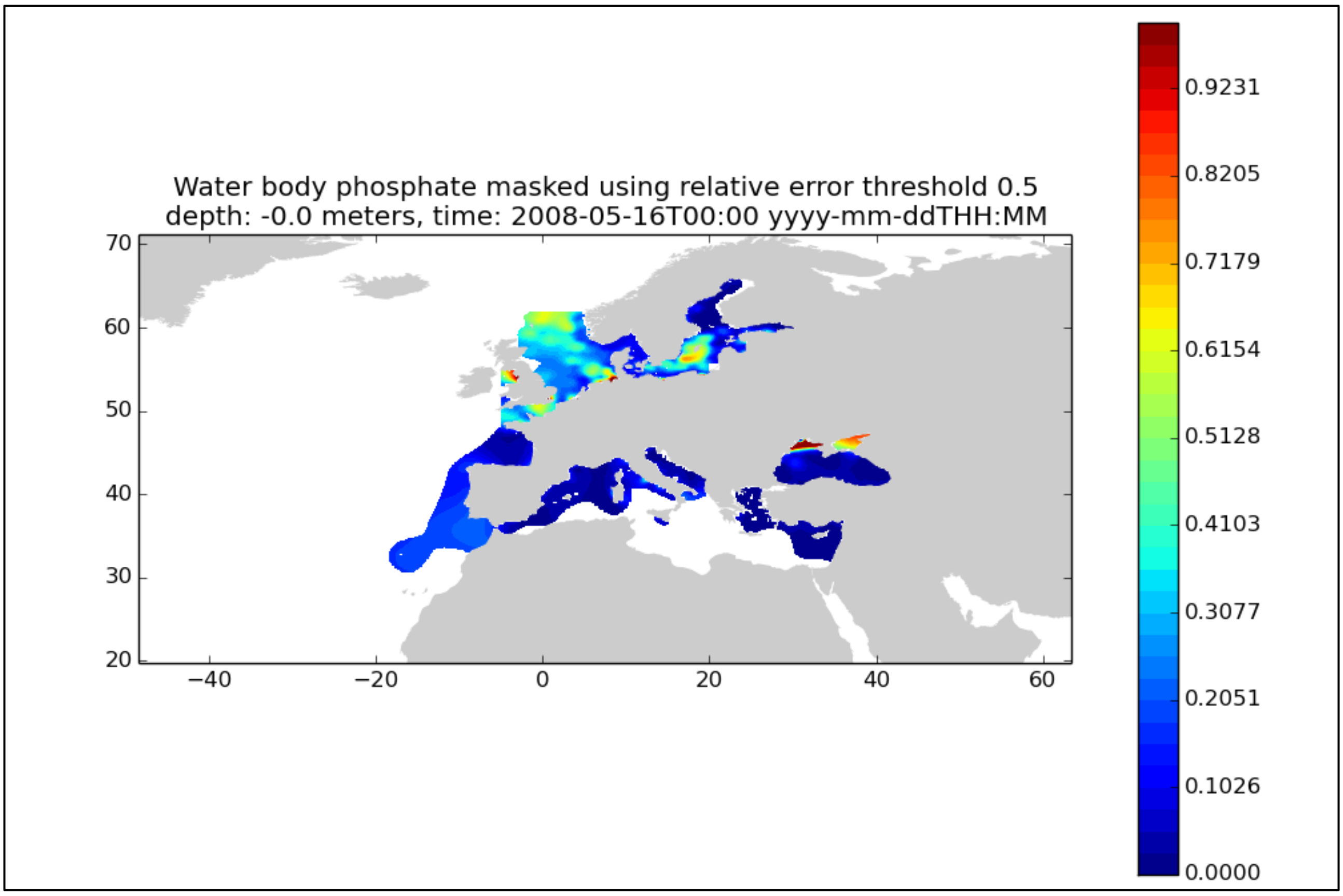

**Fig. 6: 10 years average of water body phosphate concentration (μmol-P $l^{-1}$) in the surface layer for all EU sea basins (years 2003-2013). Interpolated Maps can be selected from the Ocean Browser viewing service interface with the following steps: choose "Select data products" button, scroll and choose in the pop-up window from the list of available products and then select "Add layer" in the lower left corner of the pop-up window. Maps can also be downloaded in different formats obtaining results as in this example (PNG file).**

Profiles and time series plots are automatically generated from the Regional aggregated and validated datasets (called Regional buffers), thanks to a service bases on WPS OGC standard, and can be dynamically customized. (Fig.7)

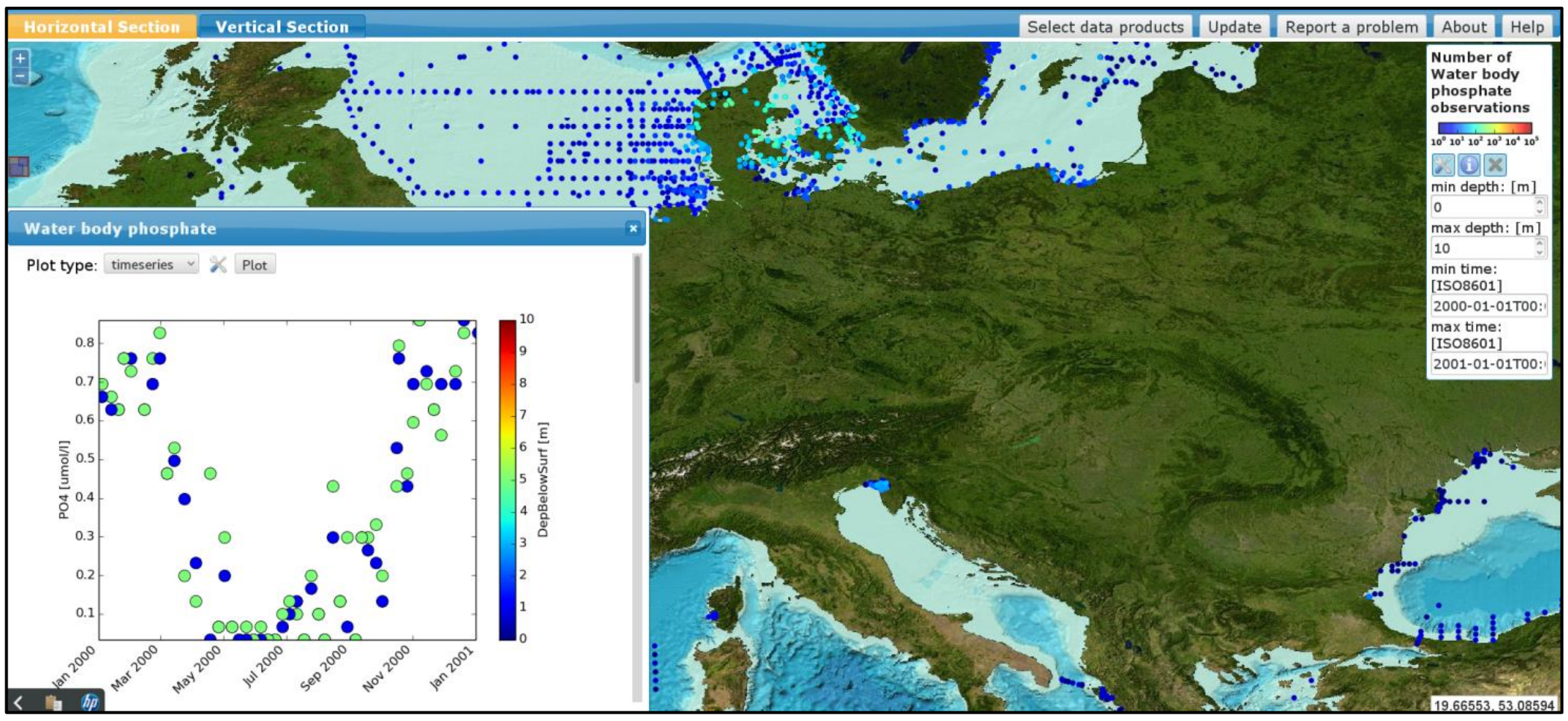

**Fig. 7: Screenshot from the web portal showing the Time Series dynamically plotted and visualized thanks to the OGC WPS services**

There are ongoing efforts to develop a more efficient information management thanks to a system of data buffers hosted in a cloud system. Data are harvested and validated in buffers and are then used for product generation.

## 7. Conclusions and perspectives

EMODnet is a long term marine data initiative developed through a stepwise approach aiming to ensure that European marine data will become easily accessible, interoperable and free of restrictions on use. EMODnet Chemistry started in 2009 to fulfil EU Marine Strategy Framework Directive requirements for the assessment of eutrophication and contaminants, following EU INSPIRE Directive rules.

With the start of EMODnet phase II, DG MARE and DG ENV started a coordination table to agree on a joint process and to identify how EMODnet can best contribute in practical terms to the MSFD. EMODnet Chemistry implemented a set of recommendations, in communication with regional sea conventions (RSC) contracting parties. The situation is not homogeneous in EU sea basins. While much of the chemistry and contaminant data are well organized within OSPAR Commission and Helsinki Convention (HELCOM), namely in the North and Baltic sea respectively, EMODnet Chemistry has a more useful role in the Mediterranean where these outputs are less well organized. A Memorandum of Understanding with the Commission on the Protection of the Black Sea against Pollution (Bucarest Convention) is under preparation to formalize the cooperation in terms of providing dedicated access to EMODnet Chemistry regional products for supporting management of MSFD indicators as well as increasing participation in the Advisory Groups meetings. A similar step is under discussion with the Information and Communication Regional Activity Center (INFO-RAC) through the United Nations Environmental Programme, Coordinating Unit for the Mediterranean Action Plan for the Barcelona Convention (UNEP/MAP).

These on-going efforts show the importance of EMODnet Chemistry results and the extensions that might be planned in view of the last EMODnet implementation phase aiming at a full resolution.

In the next years, EMODnet Chemistry could play an important role in the European environmental reporting landscape with two main tasks. The first task consists in providing standardized and quality checked buffers of data for specific Regions. The second task is to act as an umbrella providing standards, best practices and infrastructure to aggregate at Regional level the single member states.

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
