# Peer review of "The role of EMODnet Chemistry in the European challenge for 2 Good Environmental Status"

_Natural Hazards and Earth System Sciences, 2016_

## Referee Comment (RC1) · Anonymous Referee #1 · 17 Aug 2016

Summary: The author presents EMODNet-Chemistry, i.e., a portal that should deliver data and metadata for different chemical groups that are directly related to the Marine Strategy Framework Directive and, in particular, to the definition and monitoring of the Good Environmental Status of marine environments. The paper also provides a good and clear explanations of the main objectives of the EU in addressing maritime issues and describes relationships and roles among different European "structures" such as the MSFD, the DG-MARE, etc.

General Comment: The paper is well structured and it clarifies several "entities", efforts, and tools that the EC and the EU set in order to bridge the gaps among marine science, data collection and availability, and marine policies. The reader will definitely appreciate the clear explanation of how the EMODNet-Chemistry portal (as well as similar portals) addresses the main environmental protection needs. However, there

are several, specific points that make the paper not suitable for a scientific audience: some parts need to be expanded (the authors may not be familiar with some definitions) and most of the figures are not well presented (see specific comments). In particular, most of the figures are not mentioned within the text.

Specific comments: Line, 9: The authors should provide a very brief explanation of the GES... several non-EU marine scientists will not understand what the Good Environmental Status is.

Line 27: Define here the acronyms GES

Line 28-29: This sentence is good for the abstract. Here, in Intro, the authors should spend some more words for defining and explaining the GES.

Line 35: rephrase as: "Some efforts have already been undertaken by Member States in 2012, which provided initial assessments of...[explain]"

Line 40: rephrase as: "As a consequence, evaluation at higher level..."

Line 42-43: This sentence is not clear. Please, explain better.

Line 50: A reference is needed here.

Line 65: Please, rephrase as: "Metadata, i.e., all the information needed to describe exhaustively the data, are..."

Line 70: Please, rephrase as: "The standardization is done at two main levels by following the interoperability principles provided by INSPIRE: "

Line 72: Replace "thanks to" with "by means of"

Line 90: The list should end with a come, such as "seabed habitats, and physical oceanography."

Line 100: I would write "Ocean Data View (ODV)". Some readers may not know what ODV is.

[Figure]

Line 109; Rephrase as "in data collection, data analyses, validation, and creation. . ."

Line 129: Refer here to Fig. 1

Figure 1: In general all figures seem to be done in haste. In Figure 1 caption should clearly explain the meaning of colors, logos, and logo positions.

Figure 2: This figure is quite useless. It would be much more interesting to show a density map, or something similar, which might highlight the differences among sub-basins in the EMODNet-Chemistry coverage. The figure is note cited within the text.

Figure 3: The figure is not cited within the text

Table 1 and 2: They are not cited within the text. Moreover, I do not understand the text "With the following. . ." on top of these tables. It looks like that the tables want to be connected to Figure 4. This is unusual and distractive.

Line 210-213: It would be better to provide some references here. See Colella et al. (2016) [PloS one 11 (6), e0155756] and references therein.

Figure 5: Yet, this is a screen shot from the web. It would be better to show an actual figure that is downloadable from the web and then explain in the caption how that map can be obtain from the portal. The figure is not cited within the text.

Figure 6: Is this really useful? The figure is not even cited within the text. If the author believe that Fig. 5 is needed I would strongly recommend not to use a slide from power point. This is not suitable for a scientific publication.

Figure 7: I understand that here it is useful to show a screen shot. However, the caption should state this.

---

## Referee Comment (RC2) · Anonymous Referee #2 · 11 Sep 2016

Review of the manuscript: The role of EMODnet chemistry in the European challenge for good environmental status. M. Vinci, A.Giorgetti, M. Lipizer Submitted to Natural Hazards and Earth System Sciences. REf: NHESS226-2016.

The paper describes the effort carried out by the EMODnet Chemistry community to make available, to the marine sciences audience, a reliable and effectively usable dataset for chemical and biogeochemical properties for theEuropean Seas, aiming to contribute the information and knowledge base necessary for the EU-MSFD objective of "good environmental status" for the European Seas.

It is an important and highly valuable effort. The present manuscript does not attempt to extract scientific information from the data collected, but tries to provide a description of the structure and the quality of the data that EMODnet chemistry is going to make

available to the marine scientists communities.

Therefore, the manuscript, even if it cannot precisely be considered as a scientific paper providing original findings, deserves (in principle) publication. Unfortunately due to a series of formal problems it cannot be published in the present form, as it looks like hastily written, without taking much care in clarity and ordered strucure.

A revision of the formal structure of the manuscript is absolutely mandatory.

Many concepts and information are taken for granted and a general reader might therefore find the manuscript rather confusing.

Figures (see below) are not correctly referenced in the text and often the relative captions are very, very sloppy.

Below some specific remarks that I hope might hel the Authors to improve the manuscript.

Line 50: please explain better how a data management system could achieve interoperability and resillence. The explanations given are still a bit "obscure".

Line 111: MSFD and not MSDF

Line163. explain better the criteria for data restriction

Line 195 and followings. Explain better (for the general reader) the meaning of codes such as P01 ocabulary and P35 vocabulary.

Section 6. Spend more words tg illustrate the procedute for data mapping (DIVA protocol)!!!!

Section 6 Validation loop must be described better. Just putting a (not referenced) figure with a sloppy caption is not enough!!!!!

Figures and tables are not correctly referenced in the text. Please reference them correctly. Just writing (for instance) "with the following data policy distribution" and

putting below a figure (or a table) is not OK. Moreover figure captions need to be rewritten in order to be more consistent with the pertinent text and with the figures themselves

In particular:

Figs 1, 2, 3, 5, 6, 7 are not referenced in the text Tab 1. Is not referenced in the text and the caption needs rewrtiting Fig. 4 needs a better caption.

---

## Author Comment (AC1) · 28 Oct 2016

Anonymous Referee #1 Summary: The author presents EMODNet-Chemistry, i.e., a portal that should deliver data and metadata for different chemical groups that are directly related to the Marine Strategy Framework Directive and, in particular, to the definition and monitoring of the Good Environmental Status of marine environments. The paper also provides a good and clear explanations of the main objectives of the EU in addressing maritime issues and describes relationships and roles among different European "structures" such as the MSFD, the DG-MARE, etc. General Comment: The paper is well structured and it clarifies sev-

eral "entities", efforts, and tools that the EC and the EU set in order to bridge the gaps among marine science, data collection and availability, and marine policies. The reader will definitely appreciate the clear explanation of how the EMODNet-Chemistry portal (as well as similar portals) addresses the main environmental protection needs. However, there are several, specific points that make the paper not suitable for a scientific audience: some parts need to be expanded (the authors may not be familiar with some definitions) and most of the figures are not well presented (see specific comments). In particular, most of the figures are not mentioned within the text.

We have considered carefully reviewer #1's observations and suggestions. The manuscript and the figures were re-organized, integrating the text with more explanations on the concepts presented and providing additional references. As suggested, several parts were expanded, all needed figures and tables were introduced and cited in the text, and figure quality was improved. The detailed answers to the comments are given below, after the Reviewer's indications. A reviewed version of the manuscript will be included in the reply supplement to clarify the improvements. In red are highlighted the changes done following the comment n.1 while in orange the changes done following the comment n.2.

Specific comments:

Line, 9: The authors should provide a very brief explanation of the GES: : : several non-EU marine scientists will not understand what the Good Environmental Status is.

Line 27: Define here the acronyms GES

Line 28-29: This sentence is good for the abstract. Here, in Intro, the authors should spend some more words for defining and explaining the GES.

The text has been integrated as follows: The European Union has set the ambitious objective to reach within 2020 the goal of Good Environmental Status (GES) for our oceans and seas. The challenge consists in facing the environmental degradation

caused by years of unsustainable and inefficient growth model. The Marine Strategy Framework Directive (MSFD, European Commission 2008) adopted in 2008, with its eleven descriptors and related indicators, represents the legislative framework and the backbone of this work. MSFD defines the GES in Article 3 as: "The environmental status of marine waters where these provide ecologically diverse and dynamic oceans and seas which are clean, healthy and productive". GES means that the different uses made of the marine resources are conducted at a sustainable level, ensuring their continuity for future generations. Going more in detail for the restoration and the safeguard of this status, the ecosystems (including their hydro-morphological, physical and chemical conditions) should be fully functioning and resilient to human-induced environmental change, the decline of biodiversity caused by human activities should be prevented and biodiversity is protected. The human activities (introducing substances and energy) shouldn't cause pollution effects and noise from human activities should be compatible with the marine environment and its ecosystems.

Line 35: rephrase as: "Some efforts have already been undertaken by Member States in 2012, which provided initial assessments of: : :[explain]"

We suggest the following improvement: Some efforts have already been undertaken by European Member States which provided in 2012 the initial assessment on the state of the environment of the national marine waters. This assessment reported on environmental status determined in a holistic way, according the 11 descriptors, and on the objectives and targets to reach GES following the articles 8, 9 and 10 of the MSFD. The results of the first phase allowed to recognize gaps and needs in data availability, large heterogeneity of methodological approaches to report information and spatial inconsistency within Member States regarding coastal – offshore data. These outcomes clearly indicated that more efforts are urgently needed if the EU is to reach its goal. More has to be done on the cooperation side and especially on the integration between Member States and Regional Sea Conventions (RSC). The report from the Commission on the first phase of implementation of MSFD indicates a high level of heterogeneity among Member States reports and in several cases poor data availability and accessibility (Dupont et al., 2014; Palialexis et al., 2014)

Line 40: rephrase as: "As a consequence, evaluation at higher level: : :"

As a consequence, evaluation at higher level (Regional and EU) is difficult to perform.

Line 42-43: This sentence is not clear. Please, explain better.

This first phase of MSFD implementation has somehow brought Europe one step closer to the ecosystem approach. However, the recognized gaps in data and information, the high heterogeneity in assessment approaches should guide the stakeholders involved in MSFD implementation to develop a more homogeneous approach. In view of the revision of the assessment in 2018, several efforts are required to overcome the shortcomings identified in the first reporting phase. Going more in detail, the actions should be focused on different aspects like: revised criteria for GES, methodological standards and standardised methods for monitoring, assessment and data availability, implementation of integrated information systems at regional and EU level.

Line 50: A reference is needed here.

In the field of marine research, during the last decades several oceanographic data management initiatives faced the challenge of data availability, interoperability and resilience at Pan-European level. (EU MAST MTP II MATER 1996-1999, EU MAST-INCO MEDAR 1999-2001, FP6 SeaDataNet 2006-2011, FP7 SeaDataNet2 2011-2015).

Line 65: Please, rephrase as: "Metadata, i.e., all the information needed to describe exhaustively the data, are: : :"

The simple but efficient idea was the active collection of the EU oceanographic data at national level carried out by a network of National Oceanographic Data Centres (NODCs). The collection of those data was done in direct communication with the data originators to ensure the best set of measured data and related metadata. Metadata, that are all the information needed to describe exhaustively the data, reply to a set of basic but fundamental questions like: who, where, when, what and how about the collected information. For this reason they are key elements to enable efficient browsing and discovering.

Line 70: Please, rephrase as: "The standardization is done at two main levels by following the interoperability principles provided by INSPIRE: "

The standardization is done at two main levels by following the interoperability principles provided by INSPIRE: syntactic and semantic. The first is done providing common formats for the files providing metadata and data (XML ISO, ascii).

Line 72: Replace "thanks to" with "by means of"

The second is done by means of a set of common vocabularies that let to "use the same language" to describe data and metadata over time, different projects and nationality.

Line 90: The list should end with a come, such as "seabed habitats, and physical oceanography."

A pilot project was launched by DG-MARE in 2009 to create the components of the European Marine Observation and Data Network (so called ur-EMODnet), as proposed in the EU Green Paper on Future Maritime Policy (European Commission 2006), consisting in six thematic data portals managing data on bathymetry, marine geology, chemistry, biology, seabed habitats, and physical oceanography.

Line 100: I would write "Ocean Data View (ODV)". Some readers may not know what ODV is.

• Standard data transport formats Ocean Data View (ODV) ASCII, MEDATLAS and NetCDF (CF);

Line 109; Rephrase as "in data collection, data analyses, validation, and creation: : :"

The partnership involved a subgroup of the SeaDataNet network of National Oceanographic Data Centres (NODCs) with specific experience in data collection, in data analyses, validation, and creation of products and in the technical partners who further developed SDN infrastructure.

Line 129: Refer here to Fig. 1

The new phase includes data collection for all European sea-basins: the Baltic Sea, the North-East Atlantic Ocean, the Mediterranean Sea and the Black Sea and involves 46 partners (Fig.1), both from research institutes and national monitoring agencies.

Figure 1: In general all figures seem to be done in haste. In Figure 1 caption should clearly explain the meaning of colors, logos, and logo positions.

We suggest as follow:

Figure 1: Geographic coverage of EMODnet Chemistry partnership. Logos indicate the nationality of the partner institutes.

Figure 2: This figure is quite useless. It would be much more interesting to show a density map, or something similar, which might highlight the differences among subbasins in the EMODNet-Chemistry coverage. The figure is note cited within the text.

We suggest the following improvement:

Fig. 2: Density and distribution of water body phosphate harvested stations

Figure 3: The figure is not cited within the text

We suggest to change the position of the figure (now figure 4) and table and cite them as follow:

By maximizing the availability of data to a larger community, SeaDataNet promotes the use of these data, thereby ensuring that their maximum value can be realized and thus contribute to increase knowledge of the marine environment. Fig. 4 shows temporal distribution of nutrient data, spanning from 1900 to 2016; table 1 shows the number of stations for parameters with more data available.

Fig. 4: Temporal distribution of nutrient data, spanning from 1900 to 2016 (counting 90 profiles in the current year; updates May 2016).

Table 1 number of stations for the parameters with more data available

Table 1 and 2: They are not cited within the text. Moreover, I do not understand the text "With the following: : :" on top of these tables. It looks like that the tables want to be connected to Figure 4. This is unusual and distractive.

Our suggestions are to eliminate "With the following: : :". Table one has been cited (see the previous point) while Table2 doesn't exist.

Fig.4: Temporal distribution of nutrient data, spanning from 1900 to 2016 (counting 90 profiles in the current year; updates May 2016).

Table 1: number of stations for the parameters with more data available

Line 210-213: It would be better to provide some references here. See Colella et al. (2016) [PloS one 11 (6), e0155756] and references therein.

To improve and harmonize the quality control procedures and standards adopted (at least at regional level), a quality control survey has been carried out within EMODnet Chemistry community, in order to collect the best practices in data validation and highlight gaps of the different institutes involved (Vinci et al., 2015).

See Colella et al. (2016) [PloS one 11 (6), e0155756] and references therein

Our proposal is the following: In order to accomplish the Marine Strategy Framework Directive requirements, EMODnet Chemistry developed products suitable to visualise the time evolution of a selected group of measurements and to calculate spatially distributed data products specifically relevant for MSFD descriptor 5 (eutrophication), 8 (chemical pollution) and 9 (contaminants in seafood) as typically done with satellite data (Colella et al., 2016; Gohin et al., 2008)

Figure 5: Yet, this is a screen shot from the web. It would be better to show an actual figure that is downloadable from the web and then explain in the caption how that map can be obtain from the portal. The figure is not cited within the text.

The figure will be modified and will be cited in the text as follows:

Interpolated maps are now generated, mainly for nutrients, with 10 years moving window in order to find a balance between the duration of the environmental evaluation cycle for Member States (to provide maps with a time frame near to the 6 years process of the Member States evaluation) and the number of years that guarantee a sufficient data coverage. An example of a visualization useful for the assessment of eutrophication and in particular of nutrient concentration in the water column is presented in Fig. 6, which displays surface distribution of phosphate concentration in spring for the decade 2003-2013 (centered in 2008)..

Fig. 6: 10 years average of water body phosphate concentration ($\mu$mol-P l-1) in the surface layer for all EU sea basins (years 2003-2013). Interpolated Maps can be selected from the Ocean Browser viewing service interface with the following steps: choose "Select data products" button, scroll and choose in the pop-up window from the list of available products and then select "Add layer" in the lower left corner of the pop-up window. Maps can also be downloaded in different formats obtaining results as in this example (PNG file).

Figure 6: Is this really useful? The figure is not even cited within the text. If the author believe that Fig. 5 is needed I would strongly recommend not to use a slide from power point. This is not suitable for a scientific publication.

We suggest the following improvement with improved figure and citation in text: Figure 6 has been moved up in the "Data quality" section and is now Fig.5. The figure quality has been improved and it is the cited in the text as follows:

... The main goal of this activity is to obtain a harmonized dataset (e.g. a unique dataset of phosphate concentration in the water column starting from different datasets of phosphate concentration expressed with different units) that could be used to generate homogeneous data products. The results of the regional quality control are sent to the data collators (NODCs) to correct errors or anomalies in the original copy of the data available in the EMODnet infrastructure. This feedback loop guarantees data quality upgrade (Fig.5).

Fig. 5: Data validation loop

To improve and homogenize the quality control procedures and standards adopted (at least at regional level), a quality control survey has been carried out within EMODnet Chemistry community, in order to collect the best practices in data validation and highlight gaps of the different institutes involved (Vinci et al., 2015).

Figure 7: I understand that here it is useful to show a screen shot. However, the caption should state this. We suggest to improve the caption and cite the figure as follow:

Profiles and time series plots are automatically generated from the Regional aggregated and validated datasets (called Regional buffers), thanks to a service bases on WPS OGC standard, and can be dynamically customized. (Fig.7)

Fig. 7: screenshot from the web portal showing the Time Series dynamically plotted and visualized thanks to the OGC WPS services

Please also note the supplement to this comment:
http://www.nat-hazards-earth-syst-sci-discuss.net/nhess-2016-226/nhess-2016-226-AC1-supplement.pdf

**Fig. 1.**

[Figure]

**Fig. 2.**

[Figure]

**Fig. 3.**

**number of stations per year**

**Fig. 4.**

Data are collected, checked, flagged and completed with metadata by National Collators

Questionnaire based on ISO/IEC 17025:2005

+

Update the official copy of data data extraction

Feedback loop for data quality upgrade

Regional data buffers

Report errors to the data originators

QCd data buffers

Data aggregation data QC with ODV Correlation of params

**Fig. 5.**

[Figure 6 full-page scientific map image]

**Fig. 6.**

[Figure]

**Fig. 7.**

**Supplement:**

[revised manuscript text omitted]

---

## Author Comment (AC2) · 28 Oct 2016

Anonymous Referee #2 Review of the manuscript: The role of EMODnet chemistry in the European challenge for good environmental status. M. Vinci, A.Giorgetti, M. Lipizer Submitted to Natural Hazards and Earth System Sciences. REf: NHESS226-2016.

The paper describes the effort carried out by the EMODnet Chemistry community to make available, to the marine sciences audience, a reliable and effectively usable dataset for chemical and biogeochemical properties for theEuropean Seas, aiming to contribute the information and knowledge base necessary for the EU-MSFD objective of "good environmental status" for the European Seas. It is an important and highly valuable effort. The present manuscript does not attempt to extract scientific information from the data collected, but tries to provide a description of the structure and the quality of the data that EMODnet chemistry is going to make available to the marine scientists communities. Therefore, the manuscript, even if it cannot precisely be considered as a scientific paper providing original findings, deserves (in principle) publication. Unfortunately due to a series of formal problems it cannot be published in the present form, as it looks like hastily written, without taking much care in clarity and ordered strucure. A revision of the formal structure of the manuscript is absolutely mandatory. Many concepts and information are taken for granted and a general reader might there- fore find the manuscript rather confusing. Figures (see below) are not correctly referenced in the text and often the relative cap- tions are very, very sloppy.

The authors acknowledge the comments and suggestions of the reviewer and the need of a revision of the manuscript with the objective to improve the clarity of the described topic. The manuscript has been integrated explaining better the concepts (that some-times were taken for granted). Presentation of figures and tables were improved following the reviewer's suggestions. Following these actions we believe that the manuscript has already improved and been made clearer. If the editor considers it necessary, we can also reorganize the structure of text, add a glossary to facilitate understanding of several specific terms used in the manuscript and the text can be revised by a native English speaker. Below the replies to the specific comments received. A reviewed version of the manuscript will be included in the reply supplement to clarify the im-provements. In red are highlighted the changes done following the comment n.1 while in orange the changes done following the comment n.2.

Below some specific remarks that I hope might help the Authors to improve the manuscript.

Line 50: please explain better how a data management system could achieve interop-erability and resillence. The explanations given are still a bit "obscure".

We suggest the following improvement.

In the field of marine research, during the last decades several oceanographic data management initiatives faced the challenge of data availability, interoperability and resilience at Pan-European level. (EU MAST MTP II MATER 1996-1999, EU MAST-INCO MEDAR 1999-2001, FP6 SeaDataNet 2006-2011, FP7 SeaDataNet2 2011-2015). Interoperability is defined as "the ability of a system to work with or use the parts of another system", while resilience is defined as "the ability of a system to cope with change". The translation of these principles in the oceanographic data management consists in the development of a long life system able to easily interact with other systems. As example the adoption of common formats for data and metadata and a system of common vocabularies ensure that the network of involved persons is working in a homogeneous environment from the syntactic and semantic point of view (speaking a common language). The resilience is safeguarded by metadata and quality flags that provide clear knowledge of which kind of information the users are handling even long time after the data measurement (e.g. use of historical data for time series studies).

Line 111: MSFD and not MSDF

Ok

Line163. explain better the criteria for data restriction

Following the comment to the manuscript received by the previous reviewer this part has been eliminated. The text describing the data policies is available from line 175 until 186 as follow:

Data access is regulated by a data policy (defined in agreement with data originators) which aims to establish a balance between the right of the originator to get proper acknowledgment for data acquisition, and the need for open access through free and unrestricted exchange of data, meta-data and data products. The analysis of data policies Ffor EMODnet Chemistry data shows differences between data access restrictions for nutrients and contaminants (Fig. 3).

Fig. 3: Data policy for nutrients and contaminant data

Data requests from registered users are handled by NODCs through a data policy management system. Unrestricted data are freely available while restricted ones need negotiation with data originators. This kind of filter on data access is an effective way to establish contacts and trust between data originators and data management centres, ensuring correct acknowledgement, which ultimately encourages data sharing.

Line 195 and followings. Explain better (for the general reader) the meaning of codes such as P01 vocabulary and P35 vocabulary.

We prefer a more general explanation of the vocabularies involved in our workflow to avoid too specific or technical descriptions. Following this we suggest the following improvement.

. . . Data aggregation is done with the objective to unify the various analytic terms into a unique aggregated term with conversion to a unique measurement unit.. The ODV software has a built-in aggregation procedure applying a number of business rules like possible units conversions. (Lowry R. et al., 2013) The main goal of this activity is to obtain a harmonized dataset (e.g. a unique dataset of phosphate concentration in the water column starting from different datasets of phosphate concentration expressed with different units) that could be used to generate homogeneous data products. The results of the regional quality control are sent to the data collators (NODCs) to correct errors or anomalies in the original copy of the data available in the EMODnet infrastructure. This feedback loop guarantees data quality upgrade (Fig.5).

Section 6. Spend more words to illustrate the procedure for data mapping (DIVA protocol)!!!!

We suggest the following improvement.

The interpolated maps have been produced with the variational inverse method (VIM;

Brasseur et al., 1996), using the software DIVA (Data-Interpolating Variational Analysis; Troupin et al., 2010). DIVA is an appropriate numerical implementation of VIM suitable for oceanographic data spatial analysis as it is designed to obtain a gridded field from the availability of non-uniformly distributed observations (Barth et al., 2010; Troupin et al., 2012).

Section 6 Validation loop must be described better. Just putting a (not referenced) figure with a sloppy caption is not enough!!!!!

Figure 6 has been moved to section "5 Data Quality" where the "Validation loop" was described in a quite detailed way in the text from line 202 to 227, and the figure is now cited in the text. Now the figure is in the correct position and cited to link the description to the image (that in our opinion clarify in a simple but efficient way the workflow).

5. Data Quality The quality of the data is a key issue when merging heterogeneous data coming from different sources, periods and geographic areas. Within EMODnet chemistry community, commonly agreed and standardized data quality control (QC) protocols have been defined (Holdsworth, 2010) to guarantee consistency among comprehensive databases which include data from different and/or unknown origin and covering long time periods. As a first step, the data are checked and completed by collators with a standard set of metadata that provide the basic information necessary for their long term use. Afterwards, data undergo a validation loop which consists in several validation steps. The first is done by data collators, prior to the inclusion in the decentralized infrastructure and the second step, which consists in regional quality control, is performed at regional scale on aggregated datasets. The first quality controls (QC) ensure that position and time of data are realistic and compare measurements with broad ranges and specific regional ranges. Whenever available, data are also compared with climatology. As a result of the first QC step, all data are archived with a quality flag value that provides information about their reliability. At this point, data aggregation and regional quality control are performed at regional scale, following a common protocol. Data aggregation is done with the objective to unify the various analytic terms into a unique aggregated term with conversion to a unique measurement unit. The ODV software has a built-in aggregation procedure applying a number of business rules like possible units conversions. (Lowry R. et al., 2013) The main goal of this activity is to obtain a harmonized dataset (e.g. a unique dataset of phosphate concentration in the water column starting from different datasets of phosphate concentration expressed with different units) that could be used to generate homogeneous data products. The results of the regional quality control are sent to the data collators (NODCs) to correct errors or anomalies in the original copy of the data available in the EMODnet infrastructure. This feedback loop guarantees data quality upgrade (Fig.5). Fig. 5: Data validation loop

To improve and homogenize the quality control procedures and standards adopted (at least at regional level), a quality control survey has been carried out within EMODnet Chemistry community, in order to collect the best practices in data validation and highlight gaps of the different institutes involved (Vinci et al., 2015).

Figures and tables are not correctly referenced in the text. Please reference them correctly. Just writing (for instance) "with the following data policy distribution" and putting below a figure (or a table) is not OK. Moreover figure captions need to be rewritten in order to be more consistent with the pertinent text and with the figures themselves In particular: Figs 1, 2, 3, 5, 6, 7 are not referenced in the text Tab 1. Is not referenced in the text and the caption needs rewrtiting Fig. 4 needs a better caption.

Authors agree on the improvement of figures, tables, captions and references that have been already updated in text.

Fig.1 is cited now at line 150 Fig.2 is cited now at line 168 Fig.3 is cited now at line 178 Fig.4 and Tab.1 are cited now at line 189 Fig.5 is cited now at line 221 Fig.6 is cited now at line 242 Fig.7 is cited now at line 252

Please also note the supplement to this comment:

http://www.nat-hazards-earth-syst-sci-discuss.net/nhess-2016-226/nhess-2016-226-AC2-supplement.pdf

[Figure]

[Figure]

[Figure]

**Fig. 1.**

Questionnaire based on ISO/IEC 17025:2005

+

Data are collected, checked, flagged and completed with metadata by National Collators

Update the official copy of data data extraction

Feedback loop for data quality upgrade

Regional data buffers

Report errors to the data originators

QCd data buffers

Data aggregation data QC with ODV Correlation of params

**Fig. 2.**

**Supplement:**

[revised manuscript text omitted]